# In-Vitro Characterization of mCerulean3_mRuby3 as a Novel FRET Pair with Favorable Bleed-Through Characteristics

**DOI:** 10.3390/bios9010033

**Published:** 2019-02-28

**Authors:** Kira Erismann-Ebner, Anne Marowsky, Michael Arand

**Affiliations:** Institute of Pharmacology and Toxicology, University of Zurich, 8057 Zurich, Switzerland; kira.erismann-ebner@pharma.uzh.ch (K.E.-E.); marowsky@pharma.uzh.ch (A.M.)

**Keywords:** 3D fluorescence spectra, FAMPIR, fluorescent protein, excitation

## Abstract

In previous studies, we encountered substantial problems using the CFP_YFP Förster resonance energy transfer (FRET) pair to analyze protein proximity in the endoplasmic reticulum of live cells. Bleed-through of the donor emission into the FRET channel and overlap of the FRET emission wavelength with highly variable cellular autofluorescence significantly compromised the sensitivity of our analyses. Here, we propose mCerulean3 and mRuby3 as a new FRET pair to potentially overcome these problems. Fusion of the two partners with a trypsin-cleavable linker allowed the direct comparison of the FRET signal characteristics of the associated partners with those of the completely dissociated partners. We compared our new FRET pair with the canonical CFP_YFP and the more recent mClover3_mRuby3 pairs and found that, despite a lower total FRET signal intensity, the novel pair had a significantly better signal to noise ratio due to lower donor emission bleed-through. This and the fact that the mRuby3 emission spectrum did not overlap with that of common cellular autofluorescence renders the mCerulean3_mRuby3 FRET pair a promising alternative to the common CFP_YFP FRET pair for the interaction analysis of membrane proteins in living cells.

## 1. Introduction

Förster resonance energy transfer (FRET) is the non-radiative transfer of energy from an excited donor fluorophore to a neighboring acceptor fluorophore by dipole-dipole coupling, resulting in the emission of fluorescence characteristic for the acceptor [1]. Since the development of respective fluorescent protein probes [2], FRET has become an important analytical strategy to analyze the association of target proteins in live cells [3]. A clear advantage of this procedure over others is that, with the exception of the need to express the target proteins artificially fused to the fluorescent probes, it otherwise involves minimal invasiveness. In particular in the case of membrane proteins, the manipulations required to allow the detection of molecular associations, such as the necessary careful solubilization with detergents, bare the significant danger of producing artifacts. Therefore, we have recently developed FAMPIR (FRET analysis of membrane protein interaction in the endoplasmic reticulum), a FACS-based FRET procedure to analyze the association of membrane proteins in the endoplasmic reticulum of the mammalian cell line HEK293 [4].

FAMPIR uses the canonical CFP_YFP FRET pair for the analysis of protein association. However, the properties of this combination present a couple of shortcomings that interfere with the detection of weak FRET signals. First, the bleed-through of the FRET donor CFP into the FRET detection channel is substantial (Figure 1). Second, in the range of the emission spectrum of the FRET acceptor YFP, HEK293 cells show a highly variable autofluorescence that complicates simple mathematical correction of the CFP bleed-through [5].

Recently, mRuby3 has been developed as a novel FRET acceptor [6]. Its emission spectrum, which is compared to YFP in the longer wavelength range, does not substantially overlap with the range of autofluorescence in HEK293 cells. The FRET partner described for mRuby3, mClover3, is reported to have favorable characteristics [6], yet has an excitation spectrum very close to that of mRuby3 itself, suggesting a potential problem with direct acceptor excitation during FRET analysis (Figure 1). To potentially overcome this problem, the combination of mRuby3 with CFP-like donors might be a solution. mCerulean3 is one of the most recently reported improved versions of CFP [7], and used together with mVenus, an improved YFP variant, in FRET applications. The aim of the present study was to assess the principal suitability of mCerulean3 and mRuby3 as a FRET pair, characterize its properties in terms of bleed-through characteristics and compare these to those of established FRET pairs.

## 2. Materials and Methods

### 2.1. Materials

The plasmids pNCS-mClover3 [6], pKanCMV_mClover3_mRuby3 [6] and mCerulean3-mVenus-FRET-10 (Michael Davidson Lab) were obtained from Addgene (Cambridge, MA, USA). The plasmid pCYP2J5_P2A_CPR carrying the CFP and YFP open reading frames is an in-house construct described previously [4]. Oligonucleotides were ordered from Microsynth AG (Balgach, Switzerland). Molecular biology grade enzymes were obtained from New England Biolabs (Ipswich, MA, USA). Analytical grade chemicals were purchased from various standard laboratory chemical suppliers.

### 2.2. Generation of Expression Constructs

All expression constructs for fluorescent proteins were produced using standard cloning procedures and were based on the pNCS-mClover3 vector that allows T7 RNA polymerase-driven bacterial transgene expression. We first replaced the mClover3 sequence with a polylinker ending in a C-terminal His-tag to facilitate transgene purification. Next, the open reading frames for CFP, YFP, mVenus, mClover3, mCerulean3 and mRuby3 were amplified from the above plasmids using Pfu DNA polymerase and inserted into the NdeI/EcoRI-restriction site of pNCS_His, resulting in the bacterial expression vectors pNCS_CFP_His, pNCS_YFP_His, pNCS_mVenus_His, pNCS_mClover3_His, pNCS_mCerulean3_His and pNCS_mRuby3_His. FRET fusion expression constructs were prepared by introducing the fluorescent protein coding XbaI/EcoRI fragment from pNCS_mClover3_His or pNCS_mCerulean3_His together with a synthetic oligonucleotide composed of the strands 5′-AATTCGGTGCTGCAGGTCGTAAAGGTGCAGCTGG-3′ and 5′-TACCAGCTGCACCTTTACGACCTGCAGCACCG-3′ into the XbaI/NdeI site of pNCS_mRuby3_His to yield pNCS_mClover3_mRuby3_His and pNCS_mCerulean3_mRuby3_His. Likewise, the XbaI/EcoRI fragment of pNCS_CFP_His was cloned into the XbaI/NdeI site of pNCS_YFP_His, together with the same oligonucleotide used for the above constructs, resulting in pNCS_CFP_YFP_His. Finally, the linker between the two fluorescent protein reading frames in pNCS_mCerulean3_mRuby3_His was modified by whole plasmid PCR amplification with Pfu polymerase using the primers 5′-ATGGTCTCGAGCGGCGAAGAGCTGATCGAGGAAAATATG-3′ and 5′-CGCCCCTCGAGACCATACCAGCGTTGTACAGCTCGTCCATGCCGAG-3′, followed by XhoI restriction and direct religation of the resulting amplicon to obtain pNCS_mCerulean3_sl_mRuby3_His. Correctness of all sequences was verified by DNA sequencing. The sequences of all expression constructs described herein are provided in the Appendix A.

### 2.3. Protein Production and Purification

His-tagged fluorescent proteins from the constructs described above were expressed in *Escherichia coli* BL21-AI. Cultures of bacteria carrying any of the above plasmids were grown in LB medium in a volume of 250 mL at 37 °C and 225 rpm to an OD_600_ of approximately 0.5. Recombinant protein expression was fostered by the addition of arabinose to 100 µM, a concentration that produced favorable results in pilot experiments with all constructs, and protein expression was allowed overnight at 30 °C. The next morning, bacteria were harvested by centrifugation at 3000× *g* and 4 °C for 15 min in a swing out rotor and the pellets were resuspended in 10 mL buffer A (sodium phosphate, 20 mM, sodium chloride, 500 mM, imidazole, 20 mM, pH 7.4). Bacteria were lysed by a single pass through a FrenchPress pressuring device (SLM Instruments, Inc., Urbana, IL, USA) at a pressure setting of 3000 bar. Lysates were centrifuged for 5 min at 13,000× *g* and 4 °C in an Eppendorf benchtop centrifuge and the supernatant was subsequently purified over a 1 mL HisTrap FF column (GE Healthcare, Uppsala, Sweden) using a Model “11” Plus syringe pump (Harvard Apparatus, Holliston, MA, USA) equipped with disposable plastic syringes. The column was first equilibrated with 5 mL buffer A at a flow rate of 1 mL per min. Subsequently, the sample was applied with a flow rate of 0.5 mL per min. The column was washed with 15 mL buffer A at a flow rate of 1 mL per min, and the fluorescent protein was subsequently eluted with buffer B (sodium phosphate, 20 mM, sodium chloride, 500 mM, imidazole, 220 mM, pH 7.4) at a flow rate of 0.3 mL per min. The elution could be monitored by eye, due to the intense color of the target proteins. Elution volume was typically between 2 and 3 mL, with the majority of target protein eluting in a volume less than 1 mL. Up to 300 µL of the peak fraction was desalted by size exclusion chromatography over a prepacked Sephadex G-50 column (gel bed dimensions 0.9 × 2.0 cm) equilibrated with buffer C (Tris-HCl, 20 mM, NaCl, 100 mM, pH 7.4). After the sample had completely entered into the gel bed, buffer C was added to a combined volume (sample plus buffer) of 500 µL. Thereafter, the desalted protein was recovered by elution with 400 µL buffer C. The protein solutions were stored at −80 °C in aliquots to avoid multiple freeze thaw cycles. Protein concentrations of the final fractions were determined with a Nanodrop 1000 spectrophotometer (Thermo Fisher Scientific, Wilmington, DE, USA), using the calculated extinction coefficient based on their primary structure, assuming reduced thiol side chains. Absorption spectra of the purified recombinant proteins were recorded in quadruplicates using the same device.

### 2.4. Recording of Fluorescence Spectra

Fluorescence recordings were performed on a Tecan Infinite M200 PRO microplate reader (Tecan Sales Switzerland AG, Männedorf, Switzerland) using the associated i-control software. For plain excitation or emission wavelength scans, the built-in routine of the apparatus was used. In a typical experiment, 1–2 µg of fluorescent protein were analyzed in a volume of 200 µL buffer (Tris-HCl, 20 mM, NaCl 100 mM, pH 7.4) in a 96-well blackwell plate (clear plates work almost equally well in the visible light range but produce substantial background fluorescence in the UV range). Because of the resolution limits dictated by the fixed bandwith of 10 nm for the excitation and 20 nm for the emission wavelength, scans were performed in 5 nm steps, and care was taken that the minimum distance between excitation and emission wavelength was kept at 35 nm. The gain was typically set between 70 and 80. For the recording of trypsin digestion kinetics, the kinetic cycle command was used, cycle number was set to 20 and the kinetic interval was set to 1 min. After an initial emission scan using the appropriate settings for the analyzed FRET fusion protein to ensure a proper sensitivity range for the analysis, the proteolytic scission of the FRET partners was initiated by addition of 0.2 µg of trypsin for 2 µg of fusion protein and immediate recording of the FRET decay with the above kinetic recording program at ambient temperature.

For the recording of three-dimensional (3D) fluorescence spectra, a method was programmed to connect sequential emission wavelength scans with a serial increase in the excitation wavelength by 5 nm. To overcome the software-based limitation of the maximum of ten such steps per recording, a “new” plate was defined in the program after 10 scan cycles which set the counter back to zero. Five such units allowed the coverage of a 370–600 nm excitation wavelength to 405–670 nm emission wavelength matrix in a 5 × 5 nm resolution, composed of 1457 individual data points. Using this automated procedure, the recording of such a spectrum took 30 min. A template was generated in Excel to condense the individual measurement outputs into one continuous matrix that was finally imported into MATLAB for the visualization as a surface plot. The “Rotate 3D” and “Data cursor” functions in the MATLAB Graphics window reveal the full information content of the 3D fluorescence spectra obtained this way, which unfortunately is not obvious from the printed representation of these figures (Figure 3, Figure A2 and Figure A3) in the paper. Therefore, we deposited a selection of MATLAB figures, as well as 3D spectra recordings in excel format together with a few MATLAB routines to generate figures in the Appendix A.

### 2.5. Calculation of Förster Radii and FRET Efficiency

The Förster radii for the FRET pairs were calculated as described by Wu and Brand [8], using the software module a|e [9] in MATLAB for the calculation of the overlap integrals. FRET efficiency was estimated by a modeling approach as described by Bajar and colleagues [6].

## 3. Results

### 3.1. Production of Monomeric Fluorescent Proteins and Initial Spectral Recordings

Subcloning, expression and purification of the fluorescent proteins was substantially facilitated by the unexpected constitutive expression of the transgenes in the absence of arabinose (see Appendix B, Figure A1). Typical protein concentrations obtained after purification were in the range of 5–10 mg/mL for the HisTrap eluates and 2–5 mg/mL for the size exclusion peak fractions. The latter were used for the recording of fluorescence spectra. Conventional excitation spectra recorded at fixed emission wavelengths and emission spectra taken at fixed excitation wavelengths (Figure 1) were generally in good agreement with those published on the fluorescent protein database website [10]. The spectral characteristics of the individual fluorescent proteins relevant for this study are given in Table 1.

To overcome the restriction of a fixed “counter” wavelength in these scans we recorded the fluorescence intensity over a two-dimensional matrix ranging from 370–600 nm for the excitation and from 405–670 nm for the emission wavelength as described in Materials and Methods, with the fluorescence intensity being the third dimension, very similar to a procedure we have earlier used to characterize the fluorescence properties of small molecules [12]. Although the long recording times had a moderate impact on protein stability as evidenced by a moderate loss in signal intensity after repeated 3D spectra recordings with the same sample (data available in Appendix A), the reproducibility of spectra was excellent when the slight loss in signal intensity was compensated by normalization to the maximum intensity in the given spectrum (Figure 4D and the data collection provided in Appendix A). 3D spectra were recorded for CFP, YFP, mVenus, mCerulean3, mClover3 and mRuby3 and represented as surface plots in MATLAB (Figure A2). Close inspection of the 3D spectra further substantiated our hypothesis that the combination of mCerulean3 and mRuby3 as a FRET pair may compare favourably to the use of either CFP_YFP or mClover3_mRuby3 because of an apparent much lower overlap of the donor and acceptor emission spectra. Furthermore, the similarity of CFP to mCerulean3 and of YFP to mVenus became even more evident. After normalization to the respective maxima, their spectra could be almost perfectly superimposed, which facilitated to calculate the quantum yields for our CFP and YFP variants that were not available from the literature (Table 1).

### 3.2. Production and Analysis of FRET Fusion Pairs

In order to explore and compare the FRET behavior of the above discussed pairs, we produced fusions between the FRET partners as described in Materials and Methods. We introduced a peptide linker between the fluorescent protein pairs that allowed rapid cleavage by trypsin, due to the presence of several potential cleavage sites in the linker and the adjacent sequence motifs of the fluorescent proteins (Figure 2).

After expression and purification of the fusion proteins CFP_YFP, mClover3_mRuby3 and mCerulean3_mRuby3, 3D spectra were recorded before and after at least 1 h digestion with trypsin at ambient temperature. The spectra nicely reveal a FRET signal in the undigested fusion proteins that disappears after trypsin digestion (Figure 3 and Figure A3). Inspection of the rotating spectra in MATLAB revealed the different degree of bleed-through for the individual constructs, increasing in the order mCerulean3_mRuby3, mClover3_mRuby3 to CFP_YFP (Figure A3). Furthermore, two surprising observations were made: (1) despite the rather similar architecture of the different fusion proteins, their observed FRET intensities and modeled FRET efficiencies did not correlate with their calculated Förster radii (Table 2); (2) after trypsin digest of the CFP_YFP fusion protein, no increase of donor emission was observed although the elimination of the strong FRET signal should result in a substantial dequenching effect. Potential tryptic instability of CFP can be excluded as an explanation for this latter observation, because extended further trypsin incubation did not lead to a relevant loss in the CFP fluorescence signal intensity (see Appendix A).

### 3.3. Time Course of Tryptic Digest

In order to observe the gradual resolution of the FRET signal during trypsin incubation we switched back to single excitation wavelength fluorescence emission scans that we monitored over a time course of 20 min in 1 min intervals (Figure 4). This allowed us to monitor the progress of desired linker proteolysis while keeping the exposure of the fluorescent proteins themselves to a necessary minimum.

The tryptic digest results in an almost complete eradication of the FRET component of the spectra and a compensatory increase in the donor emission intensity for all three FRET pairs in the course of the incubation period (Figure 4A). However, the donor intensity increase in the CFP_YFP spectrum is extremely small compared to the massive loss in the associated FRET signal intensity. On the first sight, the most prominent FRET signal was obtained with the CFP_YFP pair. However, this is partly explained by the fact that the signal rides on the substantial background of donor emission. Quantification of the results revealed highest dequenching of donor fluorescence and strongest FRET signal reduction in relation to remaining background for the mCerulean3_mRuby3 pair, followed by the mClover3_mRuby3 pair, followed by the CFP_YFP pair (Figure 4B). The time course analysis revealing the half-lives of the fusion proteins of < 4 min during trypsin digestion (Figure 4C) and the control experiments (Figure 4D) finally indicate that the proteolytic cleavage is essentially complete after 20 min and the reproducibility of spectra is very high.

## 4. Discussion

In the present report we describe a rapid approach to evaluate FRET pair performance in vitro using selective proteolytic cleavage of the peptide linker between FRET donor-acceptor pairs and apply it to the assessment of the novel combination mCerulean3_mRuby3 in comparison to the classical CFP_YFP and the more recently established mClover3_mRuby3 pair. Our experiments indicate a superiority of the Cerulean_Ruby pair in terms of signal to noise ratio at the level of pure proteins over both, the CFP_YFP and the Clover_Ruby FRET pair. The experimental procedure itself is characterized by high sensitivity and very good reproducibility of the spectral recordings.

The 3D spectra of the individual fluorescent proteins and FRET fusion pairs are a particularly valuable outcome of the present study and allow in depth analysis of the excitation-emission matrix of these proteins. One definite advantage is that the stoichometry of the donor and acceptor fluorescence intensities is determined simultaneously in a single recording step and, given a sufficient stability of the fluorescent proteins during the analytic procedure, facilitates quantitative analysis. The stability can easily be assessed by repeated recordings of the same sample (for some examples see Appendix A) and proved to be sufficient for the FRET pairs used in the present study (less than 20% decay over a period of > 3 h).

One potential problem of our design is the recently reported photochromic behavior of mRuby3 [15]. Indeed, we realized some slight initial inconsistencies in the spectral recordings of mRuby3. However, it turned out that the behavior was stable and reproducible under defined recording conditions. It is likely that such problems are much more pronounced on microscopic applications, because continuous irradiation is essentially unavoidable in these procedures, while our microplate reader works with microsecond light pulses and the residence time of cells in the FACS analysis that we finally aim at are in the same order of magnitude, so that photochromicity is presently of moderate concern for us. However, for the sake of flexibility, a non-photochromic probe is definitely desirable and the recently developed red fluorescent protein mScarlet appears to be a favorable alternative to mRuby3 [15]. However, one should mention that mRuby3 has recently been successfully used as part of a genetically encoded voltage indicator called VARNAM to microscopically monitor neuronal membrane potential changes during action potential firing in real time [16].

One consequence, likely arising from mRuby3 photochromicity, is that the spectral characteristics of the protein in our hands deviated from those reported in the original publication [6], and therefore, we calculated a smaller Förster radius for the mClover3_mRuby3 pair than the one originally published (6.1 nm compared to 6.5 nm reported previously). Similarly, the Förster radius we calculated for mCerulean3_mRuby3 with 5.4 nm was not particularly exciting. On the other hand, the observed FRET efficiencies with our different protein pairs did not at all correlate with their calculated Förster radii, despite the very similar design; thus, we expected a similar fluorophore distance. The particularly high efficiency we found for the CFP_YFP pair might be explained by the fact that both variants were not mutated to counteract dimerization (no A206K variants), thus forcing them into close proximity may trigger a dynamic equilibrium in favour of a substantial fraction of the FRET partners forming a dimer, as proposed earlier for such non-mutated GFP-derived species [17]. A major contributor to the erroneous calculation of Förster radii is the uncertainty about the orientation factor κ^2^, which describes the relative orientation of FRET dipoles to each other and contributes as a factor in the denominator of the equation for the Förster radius calculation. Thus, Förster radius size linearly correlates with κ^2^, which can theoretically vary between 0 and 4 and is generally assumed to be 2/3 which represents random orientation. Therefore, any factor influencing this random orientation results in a deviation of the FRET behavior from what is expected by using the above generic value for κ^2^. A recent study has demonstrated that variation of κ^2^ can be very high with protein-based FRET pairs [18]. Regardless of the unexpectedly high FRET efficiency observed with the CFP_YFP pair, the mCerulean3_mRuby3 still performed better in terms of signal amplitude change (Figure 4B).

A final unexplained mystery revealed by our data sets is the unexpectedly low increase in the donor signal after trypsin digest of the CFP_YFP pair. Given the FRET efficiency of 0.71, a roughly three-fold increase in in the CFP signal intensity should result from the separation of the FRET partners; that itself is obvious from the loss in the acceptor emission. Indeed, the final spectrum after trypsin digest of the CFP_YFP fusion protein exactly matches the spectrum obtained from a 1:1 mixture of CFP and YFP (see Appendix A). Likewise, the sequential subtraction of the donor spectrum and the acceptor spectrum from the trypsin-digested CFP_YFP spectrum did not result in any FRET signal left. Because the subsequent stability of the CFP itself over several hours in the presence of trypsin was documented (see Appendix A), there is, as yet, no plausible explanation for this phenomenon.

Our chosen pair mCerulean3_mRuby3 has the smallest bleed-through of the three pairs tested, as predicted from the excitation and emission spectra (Figure 1). However, the obvious cost is a reduced spectral overlap, which results in a comparatively small Förster radius r0 because the spectral overlap is one of the important FRET pair-dependent variables that correlates positive with the size of r0. This difference in r0 (5.4 nm for mCerulean3_mRuby3 as compared to 6.1 nm for mClover3_mRuby3) theoretically results in a 30% difference in signal intensity when the distance between the fluorophores is close to r0 and slowly increases with the size of the distance to around 50% at a distance of around 10 nm, where the FRET efficiency itself drops below 5%.

An important consideration for our strategy to develop an improved FRET pair is the fact that for the analysis of a static FRET signal in living cells, the background signal is of major importance. In our previous work using CFP_YFP as the sensor pair in a FACS-based FRET approach [4], we observed that the cells presented with a highly variable individual autofluorescence in the wavelength range of the acceptor emission that substantially interfered with the mathematical correction of the donor bleed-through necessary for this FRET pair. This is of lower concern when working with FRET-based sensors that indicate a change in conditions, e.g., intracellular lactate concentration changes [19] or GPCR activation [20,21], because the individual cells serve as their own negative control in the before/after event assessment, and a 2% change in signal intensity can often be reliably identified. In contrast, a 2% variation of a static signal on a variable background is meaningless, which makes background reduction a major concern.

In conclusion, mCerulean3_mRuby3 is characterized as a promising FRET pair. The obvious next step is to test its suitability in live cells. FAMPIR expression plasmids are presently under construction to see whether we obtain the expected improvement in this procedure when exchanging CFP_YFP by mCerulean3_mRuby3.

## Figures and Tables

**Figure 1 biosensors-09-00033-f001:**
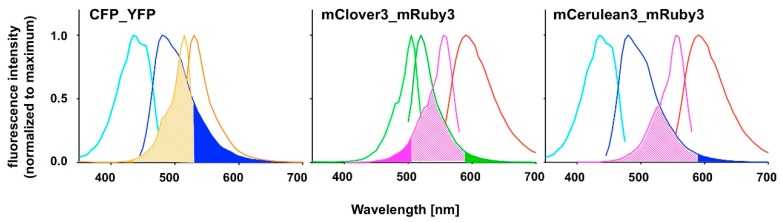
Excitation and emission spectra of fluorescent proteins presently or perspectively used as Förster resonance energy transfer (FRET) pairs. Spectra were recorded in the present study as described under Material and Methods. Spectra for CFP (light and dark blue lines), YFP (light and dark orange lines), mClover3 (light and dark green lines), mRuby3 (pink and red lines) and mCerulean3 (light and dark blue lines) are normalized to their respective maximum intensity. Overlap of donor emission spectra and acceptor excitation spectra are indicated by pale shading. Potential bleed-through problems due to donor emission (dark blue or dark green) or direct acceptor excitation (orange or pink) in the FRET channel are pointed out by color filling of the areas under the respective curves.

**Figure 2 biosensors-09-00033-f002:**
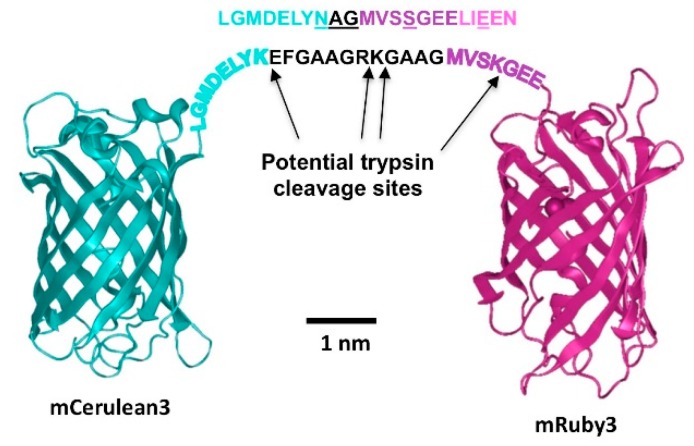
mCerulean3_mRuby3 FRET fusion construct. The proteins are represented based on the reported crystal structures of their closest relatives (5oxb for Cerulean [13], 3u0l for mRuby [14]). The amino acid sequence of the non-native peptide linker is shown in black print. The sequences in cyan represent the native Cerulean C-terminus that is flexible and thus not visible in the high resolution structure. The mRuby3 N-terminal flexible sequence is given in dark magenta. Potential trypsin cleavage sites in these sequences are indicated by the arrows. Above the fusion protein linker, the corresponding sequence that replaces it in the trypsin-resistant fusion protein is indicated. All positions where trypsin cleavage sites have been eliminated are underlined. The print size of the peptide sequences was chosen to occupy approximately the space of the amino acid sequence when stretched completely, and a scale bar based on the three-dimensional (3D) structure dimensions is included.

**Figure 3 biosensors-09-00033-f003:**
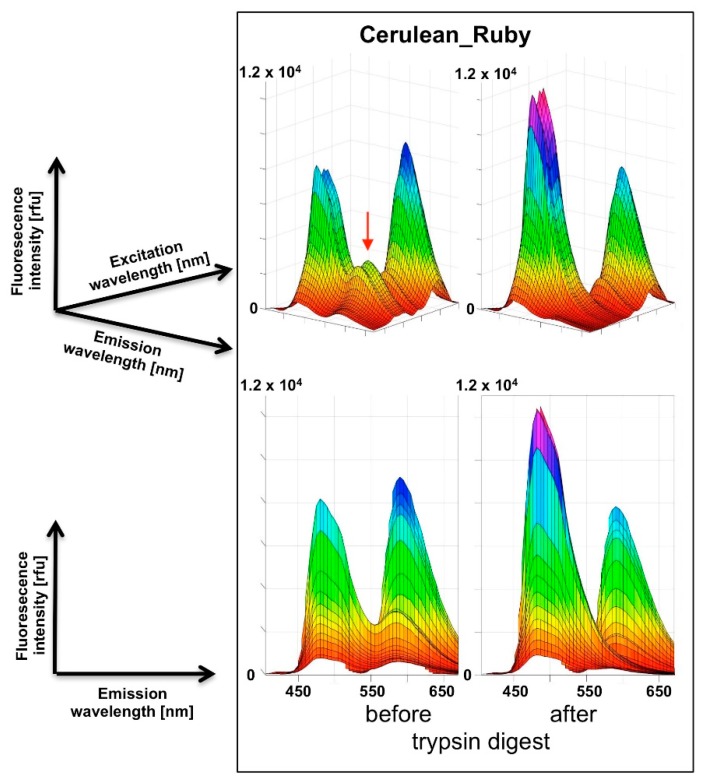
3D fluorescence spectra of the mCerulean3_mRuby3 fusion construct before and after trypsin digest. In the upper panel, the viewing angle is chosen to support the visualization of the FRET signal (red arrow) that disappears after trypsin digest, while the two-dimensional perspective in the lower panel allows the quantitative comparison of the mCerulean3 and the mRuby3 signal intensities. The mCerulean3 peak height has increased after trypsin digest, due to dequenching, while the mRuby3 peak height has moderately decreased.

**Figure 4 biosensors-09-00033-f004:**
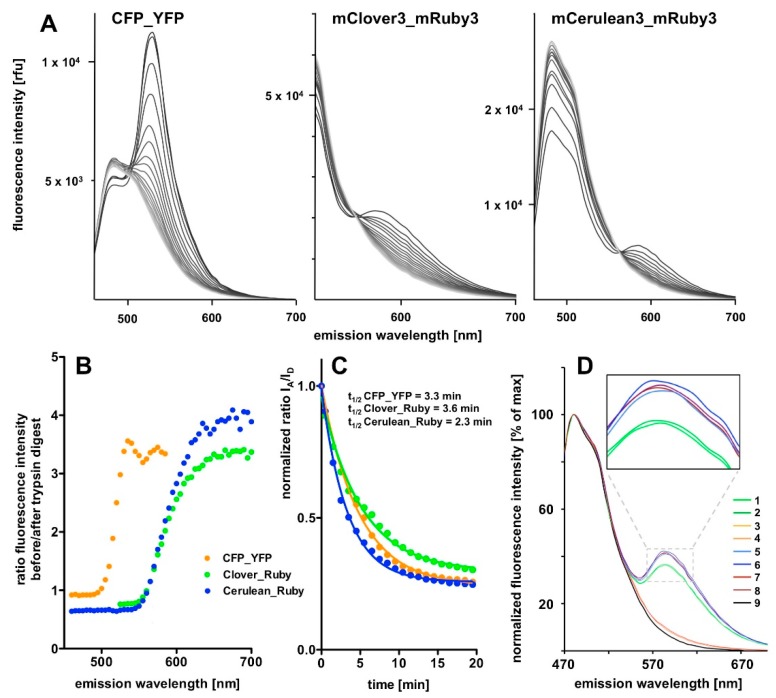
Time course of the tryptic digest of the FRET fusion proteins. (**A**) The FRET signal decay in response to tryptic digestion over time for the three FRET constructs. The excitation wavelengths were 425 nm for the recording of the CFP_YFP and the mCerulean3_mRuby3 spectra and 490 nm for the mClover3_mRuby3 spectra. FRET signals are gradually reduced over time, while the peak/intensity in the left part of the spectrum is increased. The absolute change in FRET signal decreases in the order CFP_YFP, Clover_Ruby to Cerulean_Ruby, as does the remaining residual fluorescence after the trypsin digest; the latter indicating reduced bleed-through problems. On plotting the ratio of fluorescence intensities before the start and after completion of the trypsin digest as a function of wavelength (**B**), it becomes obvious that the mCerulean3_mRuby3 pair shows the biggest amplitude. This is less evident when plotting the I_A_/I_D_ ratio over time (**C**) because the signal to noise ratio further improves with longer wavelength beyond I_A_ in case of the Cerulean_Ruby and the Clover_Ruby FRET constructs, while this is not the case for the CFP_YFP pair (**B**). (**D**) Stability and reproducibility controls: two similar experiments were performed on separate days and compared after normalization of the fluorescence intensity signals to the maximum of the respective curve. 1 and 2 (green lines): mCerulean3_mRuby3 before tryptic digest; 3 and 4 (orange lines) mCerulean3_mRuby3 after tryptic digest; 5–8: the same set of experiments with the mCerulean3_mRuby3 construct containing the short, trypsin-resistant linker (note the higher FRET efficiency of the construct and the lack of FRET signal decay after trypsin incubation (dark red lines); 9: mCerulean3 only.

**Table 1 biosensors-09-00033-t001:** Photophysical properties of the recombinant purified fluorescent proteins.

Fluorescent Protein	λ_Abs_ [nm] ^a^	λ_Em_ [nm] ^a^	ε [M^−1^ × cm^−1^] ^a^	φ*F* ^b^
CFP	436	480	37 × 10^3^	0.54^c^
mCerulean3	436	480	32 × 10^3^	0.87 [7]
mClover3	505	520	65 × 10^3^	0.78 [6]
YFP	515	530	68 × 10^3^	0.37 ^d^
mVenus	515	530	88 × 10^3^	0.64 [11]
mRuby3	559	593	79 × 10^3^	0.45 [6]

^a^ Data recorded in the present study; ^b^ data taken from the literature given in the superscript, unless stated otherwise; ^c^ calculated in this study, using mCerulean3 as the reference, ^d^ calculated in this study, using mVenus as the reference.

**Table 2 biosensors-09-00033-t002:** Characteristics of the FRET constructs.

FRET Pair	Förster Radius [nm]	FRET Efficiency
CFP_YFP	5.3	0.71
mClover3_mRuby3	6.1	0.34
mCerulean3_mRuby3	5.4	0.34

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
