# Peer review of "In-Vitro Characterization of mCerulean3_mRuby3 as a Novel FRET Pair with Favorable Bleed-Through Characteristics"

_biosensors, 2019, doi:10.3390/bios9010033_

Round 1
Reviewer 1 Report
The authors present a spectroscopic study of a novel FRET pair, mCerulean3-mRuby3, and conclude that this FRET pair has particular favorable properties. For this the authors use donor-acceptor fusion protein with a cleavable linker. Although the study is of some use, the conclusions drawn in the manuscript are not supported by the presented data, furthermore the scientific advance made is unfortunately very limited.
1. Although the greater separation of the donor and acceptor spectra reduces bleed through of the donor and direct excitation of the acceptor, this does not necessarily mean that the SN of the FRET ratio is considerably improved. Less overlap also gives less sensitized emission and donor quenching. Furthermore, the study does not provide a thorough analysis of this aspect, therefore the statement is not supported by the presented data. Bleed through or direct excitation does not only determine signal to noise of the FRET ratio, and in general is not the major concern.
2. To follow up on point one, the study is lacking a detailed analysis of Forster radii (R0) calculations for different donor-acceptor pars nor a precise quantification of the energy transfer percentages, E, based on the measured spectra.
3. The study shows a very limited set of possible donor-acceptor pairs, and there is no clear rationale for the chosen combinations. To give an example, the study by Mastop et al (2017), gives a thorough comparison of many different mTurquoise2 and acceptor pair combinations. Notably in this particular study also mRuby2 was fused to mTq2, which appeared not to be the best choice for a CFP-RFP fusion. This is not discussed in this paper. Spectrally mTq2 and mCer3, mRuby2 and mRuby3 do not differ much.
4. The choice of mCer3 over other CFP variants is not clear, for example mTq2 has been shown to be more photo-stable than mCer3, see for example Goedhart et al (2012). Older variants of Cer have been shown to be photochromic, it is not clear how photochromic mCer3 still is.
5. The authors in particular advocate the use of mRuby3, however this RFP has been shown to be highly photochromic (440/560 excitation cycles) and possesses slow and extremely variable maturation kinetics, see Bindels et al 2018 and Mastop et al 2017.
6. mRuby3 has been shown to aggregate in Golgi when studying ER-resident proteins for unknown reasons (Bindels et al 2018). The aggregation tendency gives rise to issues when membrane-FRET fusions are used, this is not favorable in cell.
7. Although the particular combination may be novel it has not been shown in this paper that an existing problem has been solved by using this combination of FPs. Moreover, due to the properties of both mCer3 and mRuby3 more detrimental experimental issues may be arise, in particular when this is used in cells. The authors argue that the spectral position of mRuby3 solves an auto fluorescence issue, however the authors do not show in this study that auto fluorescence is a big problem and that mCer3-mRuby3 solves this issue, over other FP combinations. Why should other scientists use this pair?
8. In the final discussion the authors use the Mastop paper to claim that mRuby3 solves an issue with mRuby2, however spectrally mRuby2 and mRuby3 are highly similar. The bleed through problem is very similar for mTq2-mRuby2 and mCer3-mRuby3, therefore the statement is not supported by the presented data.
9. The performance of the mCer3-mRuby3 fusion in cells cannot be extrapolated only based on this in vitro study, to draw conclusions the authors should perform experiments in cells.
10. In the introduction the authors at line 34-35 talk about association of membrane proteins in the ER, however by using FRET based techniques one has to be aware of bystander FRET, which is not linked to protein-protein association but acceptor concentration.
11. At line 124 the authors speak about the Tecan Infinite microplate reader, it is unclear if this is a true spectrophotometer. The bandwidth of 10 and 20 nm for excitation and emission respectively is rather coarse, clearly averaging out details in the spectra.
12. It is unclear if Trypsin also digests the FP barrels, has this been tested (single FPs)?
13. Based on the QY of the FPs in 5A I expect a larger increase in the donor peak. In 5B the x-axis seems to be different, only showing a small part of the donor peak.
14. The 3D plot figures (fig 2,4,6) are very difficult to read, I strongly suggest to use the presentation used in fig 5A-B, use colored lines for different excitation wavelengths.
15. In fig 5E a number of “essential controls” are shown for mCer3-mRuby3, these are however not shown for the other FRET pairs, why not?
16. The list of references is very limited for what has been published in the recent years. Although this study has some potential, it appears to be more like a pilot study that needs much more analysis and measurements in cells.
Author Response
Dear editors,
We thank the reviewers for their thoughtful comments, that substantially helped us to improve the manuscript quality.
We answer/comment in the following:
Reviewer 1:
The authors present a spectroscopic study of a novel FRET pair, mCerulean3-
mRuby3, and conclude that this FRET pair has particular favorable properties.
For this the authors use donor-acceptor fusion protein with a cleavable linker. Although the study is of some use, the conclusions drawn in the manuscript are not supported by the presented data, furthermore the scientific advance made is unfortunately very limited.
We have reworded those parts of the manuscript that may give rise to the comment that our conclusions are not supported by the presented data. Our paper clearly is a pilot study that analyzes the in vitro properties of the proposed FRET pair in comparison to others. We have also modified the title of the manuscript to point this out, in order to avoid misunderstandings
1. Although the greater separation of the donor and acceptor spectra reduces bleed through of the donor and direct excitation of the acceptor, this does not necessarily mean that the SN of the FRET ratio is considerably improved. Less overlap also gives less sensitized emission and donor quenching. Furthermore, the study does not provide a thorough analysis of this aspect, therefore the statement is not supported by the presented data. Bleed through or direct excitation does not only determine signal to noise of the
FRET ratio, and in general is not the major concern.
We have added the calculations of r0 and modeling of E. The favourable signal to noise ratio is documented in Fig. 4B. We admittedly do not present data on lack of interfering autofluorescence in the present manuscript, simply because our cell data are presently absolutely preliminary and impossible to incorporate into the present manuscript, due to the deadline for the special issue. Background in the Ruby channel is virtually inexistent , in contrast to the occasiomnally strong and highly variable autofluorescence we see in the YFP channel. One aspect of our analyses is that we perspectively intend to measure static signals that do not respond to stimuli, which means that in contrast to measurements employing FRET sensors we do not have a change in signal intensity where the starting conditions serves as the baseline for a subsequent recording of signal increase or decrease in response to a manipulation. This asks for much better controlled background signal conditions. This is now discussed more clearly in the manuscript ("Discussion").
2. To follow up on point one, the study is lacking a detailed analysis of Forster radii (R0) calculations for different donor-acceptor pars nor a precise quantification of the energy transfer percentages, E, based on the measured spectra.
See above, thanks a lot for asking for that ;)
3. The study shows a very limited set of possible donor-acceptor pairs, and there is no clear rationale for the chosen combinations. To give an example, the study by Mastop et al (2017), gives a thorough comparison of many different mTurquoise2 and acceptor pair combinations. Notably in this particular study also mRuby2 was fused to mTq2, which appeared not to be the best choice for a CFP-RFP fusion. This is not discussed in this paper. Spectrally mTq2 and mCer3, mRuby2 and mRuby3 do not differ much.
I disagree somewhat on the comment that there is no clear rational for the chosen pair: by using a red fluorescent protein, we intended to reduce donor bleed-through and interference with cellular autofluorescence, two points that are definitely met in our study. It is true that there may be many potentially better combinations out there, yet for us it looks that we have made a substantial progress with a favourable effort to benefit ratio.
4. The choice of mCer3 over other CFP variants is not clear, for example mTq2 has been shown to be more photo-stable than mCer3, see for example Goedhart et al (2012). Older variants of Cer have been shown to be photochromic, it is not clear how photochromic mCer3 still is.
In our hands, the stability of mCer3 is excellent. mTq2 may be an even better donor, and mScarlet a more suitable acceptor, but choosing mCer3_mRuby3 over CFP_YFP is certainly a (possibly first) large improvement for us.
5. The authors in particular advocate the use of mRuby3, however this RFP has been shown to be highly photochromic (440/560 excitation cycles) and possesses slow and extremely variable maturation kinetics, see Bindels et al
2018 and Mastop et al 2017.
As now discussed in the manuscript, photochromicity of Ruby may be an issue, yet a minor one for us at the moment because we do not rely on microscopic techniques. Furthermore, the recent report on the development of VARNAM indicates some usefulness for highly demanding microscopic applications of mRuby3 (Kannan et al, 2018, Nature Methods)
6. mRuby3 has been shown to aggregate in Golgi when studying ER- resident proteins for unknown reasons (Bindels et al 2018). The aggregation tendency gives rise to issues when membrane-FRET fusions are used, this is not favorable in cell.
To my understanding this was observed under conditions of OSER. We have, so far, not seen any problems with our ER resident enzyme fusions but will definitely look carefully for artifacts in our cell based assays, in order to avoid wrong conclusions
7. Although the particular combination may be novel it has not been shown in this paper that an existing problem has been solved by using this combination of FPs. Moreover, due to the properties of both mCer3 and mRuby3 more detrimental experimental issues may be arise, in particular when this is used in cells. The authors argue that the spectral position of mRuby3 solves an auto fluorescence issue, however the authors do not show in this study that auto fluorescence is a big problem and that mCer3-mRuby3
solves this issue, over other FP combinations. Why should other scientists use this pair?
I agree that the latter decision can only taken by others once we have the final data in the cellular context. However, the in-vitro analysis gives a solid basis for ourselves to take this step, this is all we intend to show in the present paper.
8. In the final discussion the authors use the Mastop paper to claim that mRuby3 solves an issue with mRuby2, however spectrally mRuby2 and mRuby3 are highly similar. The bleed through problem is very similar for mTq2- mRuby2 and mCer3-mRuby3, therefore the statement is not supported by the presented data.
This was a misinterpretation of the spectra shown in the paper and has been removed: we interpreted the spectra in Fig. 1 as excitation spectra, not absorption spectra, and thus concluded that the acceptor bleed-through would be higher than we see it with mRuby3. We do not have mRuby2 in-house to compare spectra.
9. The performance of the mCer3-mRuby3 fusion in cells cannot be extrapolated only based on this in vitro study, to draw conclusions the authors should perform experiments in cells.
Our next step.
10. In the introduction the authors at line 34-35 talk about association of membrane proteins in the ER, however by using FRET based techniques one has to be aware of bystander FRET, which is not linked to protein-protein association but acceptor concentration.
This is definitely a point that worries us a lot, one reason why we chose to use "monomeric " FRET partners. However, we do not have an indication, so far, for this bystander effect that should impress with a nonlinear relation between donor/acceptor signal intensities on the one hand and FRET intensity on the other hand (exponential increase expected)
11. At line 124 the authors speak about the Tecan Infinite microplate reader, it is unclear if this is a true spectrophotometer. The bandwidth of 10 and 20 nm for excitation and emission respectively is rather coarse, clearly averaging out details in the spectra.
This is clearly a small disadvantage of our microplate reader format, however, fluorescence emission spectra at room temperature do usually not show subtle details. Reproducibility of data was a concern in the first place that turned out to be no concern at all. The instrument produces highly reproducible spectra
12. It is unclear if Trypsin also digests the FP barrels, has this been tested
(single FPs)?
Under the conditions we use the FPs are stable for many hours (see Supplementary Materials for own assessment). The initially observed loss in signal intensity is actually more attributable to gradual adsorption to the microplate over time (in the range of much less than 10% per hour) than to proteolytic degradation. This was a particular concern with CFP, where we missed the dequenching effect, but it is definitely clear that this is not proteolytic instability in the presence of trypsin.
13. Based on the QY of the FPs in 5A I expect a larger increase in the donor peak. In 5B the x-axis seems to be different, only showing a small part of the donor peak.
We have no explanation for the lack of dequenching of CFP, as now discussed in the paper. The missing peak in the Clover spectrum (former Fig. 5B) is explained by the choice of the excitation wavelength (490 nm) which precludes to monitor emission below 425 nm. We have recorded trypsin spectra for clover_ruby also at lower excitation wavelength, yet the signal to noise ratio was less favourable.
14. The 3D plot figures (fig 2,4,6) are very difficult to read, I strongly suggest to use the presentation used in fig 5A-B, use colored lines for different excitation wavelengths.
The 3D plot are actually what we ourselves regard as the most interesting aspect of our analyses, yet I understand that the presentation on paper is not very impressive. We therefore moved the majority of spectra to the Appendix and strongly recommend to have a look at them in MATLAB.
15. In fig 5E a number of “essential controls” are shown for mCer3-mRuby3, these are however not shown for the other FRET pairs, why not?
These "essential controls" are included to give the reader a simple impression of the reproducibility of the analysis. Unfortunately, we have the trypsin-resistant variant only for the mCeru3_mRuby3 pair, but we think this is sufficient for a proof of principle. If there is lack of confidence in the quality of our data I recommend to inspect the raw data files of the 3D spectra before and after trypsin digest that are now deposited in the Supplemetary Materials
16. The list of references is very limited for what has been published in the recent years. Although this study has some potential, it appears to be more like a pilot study that needs much more analysis and measurements in cells.
It, indeed, is a pilot study, and analyses in cells will follow, yet not in the next few weeks. We have now extended the Discussion somewhat and added a few more references.
Reviewer 2 Report
In this manuscript a new CFP/YFP FRET pair, namely the mCerulean3/MRuby3 pair, is introduced and analysed with respect to its potential use as a FRET pair. In this respect the authors report on spectral properties of the involved FPs as compared to other CFP/YFP pairs. The main advantage of the new FRET pair is given by the more pronounced spectral separation of the donor emission to the acceptor emission which reduced the unwanted bleed-through in FRET measurements. Although the performed measurements and the presented results appear solid, the obtained insight from this study is on the one hand rather limited and on the other hand need to be presented more focussed.
Major comments:
General comments:
(i) In principle all figure legends are too long (in particular Fig. 4 and 5). The authors should give in the legends only descriptive information and move all detailed explanations, discussions and interpretations in the main text.
(ii) For Fig. 1 and 5 please give the corresponding FP pair naming explicit directly above the shown spectra (like in Fig. 4)
(iii) The shown 3D fluorescence spectra (Fig. 2, 4) do not give much feasible information content. The main and most important information is given in Fig. 5. Therefore, the authors should move these figures in the Appendix or in the Supporting Information.
Further comments:
Fig.1: The shown “bleed-through” and “direct excitation” contributions (shaded areas) are also dependent on (emission) band-pass or long-pass filters. The information on which filters (central wavelength, bandwidth) are considered here should be given.
Legend Fig. 2 and page 6 lines 215-217: Any discussion about relative fluorescence emission intensities between different FPs only makes sense if the concentration of the FP in the respective measurement is known or is exactly equal in all cases. As far as I can see it, absorption measurements to quantify the absolute concentration for the individual FPs were not performed.
Fig. 5: In Fig. 5E the authors investigate the performance of the different FRET constructs. Here the “ratio of fluorescence intensity” is given with respect to before and after digest. For many FRET biosensors the FRET ratio IA/ID is utilized. Could this ratio be a parameter to judge your new FP FRET-pair with respect to the others? The authors can plot this IA/ID ratio as a function of time with maximal IA/ID value before digest (largest energy transfer) and minimal IA/ID values for full digest (smallest energy transfer).
Author Response
Dear editors,
We thank the reviewers for their thoughtful comments, that substantially helped us to improve the manuscript quality.
We answer/comment in the following:
Reviewer 2:
In this manuscript a new CFP/YFP FRET pair, namely the mCerulean3/mRuby3 pair, is introduced and analysed with respect to its potential use as a FRET pair. In this respect the authors report on spectral properties of the involved FPs as compared to other CFP/YFP pairs. The main advantage of the new FRET pair is given by the more pronounced spectral separation of the donor emission to the acceptor emission which reduced the unwanted bleed-through in FRET measurements. Although the performed measurements and the presented results appear solid, the obtained insight from this study is on the one hand rather limited and on the other hand need to be presented more focussed.
Major comments:
General comments:
(i) In principle all figure legends are too long (in particular Fig. 4 and 5). The authors should give in the legends only descriptive information and move all detailed explanations, discussions and interpretations in the main text.
We have rearranged the presentation and reduced figure legends substantially
(ii) For Fig. 1 and 5 please give the corresponding FP pair naming explicit directly above the shown spectra (like in Fig. 4)
This has now been done.
(iii) The shown 3D fluorescence spectra (Fig. 2, 4) do not give much feasible information content. The main and most important information is given in Fig.
5. Therefore, the authors should move these figures in the Appendix or in the
Supporting Information.
We have largely followed this suggestion.
Further comments:
Fig.1: The shown “bleed-through” and “direct excitation” contributions (shaded areas) are also dependent on (emission) band-pass or long-pass filters. The information on which filters (central wavelength, bandwidth) are considered here should be given.
The present study was run in a microplate reader with two double monochromators that does not use bandpass filters. Once we switch to microscopic or FACS analysis these informations need to be considered, but up to now we work on the use of the microplate reader also for in cell recordings. However, we have (in our view) improved the figure by indicating spectral overlap, certainly an important point for the choice of FRET pairs
Legend Fig. 2 and page 6 lines 215-217: Any discussion about relative fluorescence emission intensities between different FPs only makes sense if the concentration of the FP in the respective measurement is known or is exactly equal in all cases. As far as I can see it, absorption measurements to quantify the absolute concentration for the individual FPs were not performed.
We have tight control over the amount of protein employed in each analyses. Quantitative data that were clearly missing have now been added to the manuscript.
Fig. 5: In Fig. 5E the authors investigate the performance of the different FRET constructs. Here the “ratio of fluorescence intensity” is given with respect to before and after digest. For many FRET biosensors the FRET ratio IA/ID is utilized. Could this ratio be a parameter to judge your new FP FRET-pair with respect to the others? The authors can plot this IA/ID ratio as a function of time with maximal IA/ID value before digest (largest energy transfer) and minimal IA/ID values for full digest (smallest energy transfer).
We have added a respective figure and discussed its usefulness. A nice side aspect of this representation turned out to be facile nonlinear regression of the fusion protein (linker) half life under trypsin incubation which is now also been added and supports the notion that the digest is essentially complete after 20 min (> 5 x t1/2).
Round 2
Reviewer 1 Report
The authors have improved the manuscript, however there are still several issues that are not adequately addressed or discussed. What really worries me is that the authors agree with my statement that this is merely a pilot study, and therefore not a matured study. My main concern is that the study is limited in the choice of donor-acceptor pairs, furthermore the extrapolation to cells has not been explored or tested at all for their particular sensor and application. Any conclusion or statement concerning cells can unfortunately not supported by data. My other concern is that without testing this particular donor-acceptor pair in vivo in cells, other scientists may think that this is the best choice, while it is known that in particular mRuby3 does not have the best and favorable properties in cells (photochromicity, extremely slow and variable maturation and clustering issues). To some extent these properties are not an issue in vitro, or otherwise stated, it is fine to perform a spectroscopic study on any pair, but one needs to be very diligent with suggestions that it would also be a good pair within a cellular in vivo context.
0) The authors added spectroscopical characteristics, however the extinction coefficients significantly differ from published values. For instance, for mRuby3 they tabulate an EC=79 M-1cm-1, whereas Lin and colleagues (their ref [6]) determined an almost twice higher value: ECs=128 M-1cm-1. If the current study is published this leads to confusion in the literature. In the discussion (line 306-308) they also underline that the spectral characteristics of mRuby3 indeed differs in their hands as compared to the original study. They attribute that to photochromism (yielding a substantially reduced Förster radius of 6.1 nm as compared to the previously published 6.5 nm) (see also point 2 and 4). Furthermore, the authors refer to figure 4B for enhanced contrast of mCerulean3-mRuby3 as compared to CFP_YFP, yet the data in fig 4B hardly shows a difference in ratio contrast between the two pairs. In addition, in figure 4A, the sensitized emission in case of CFP_YFP is clearly significantly enhanced as compared to the Clover3_mRuby3 and Cerulean3_mRuby3 pairs. So again their point of enhanced contrast is not shown by their data. (see also point 13). Of notice is that the authors also admit they do not present data on the lack of autofluorescence in cells and even state their cell data are “absolutely preliminary”.
1) Förster radii were incorporated, but the R0 values tabulated for mClover3-mRuby3 of 6.1 nm is 0.4 nm lower than the published value.
2) The answer to question 3) is not satisfactory: no simple comparison of straightforward other cyan-Red FRET pairs have been presented whereas this is a trivial effort to do so. As stated before: a much more extensive study is published elsewhere (their ref [21]) and the current paper fails to add anything to that published paper. No single convincing case can be made that mCerulean3-mRuby3 is particular beneficial within the cellular context. Not necessarily in view of the donor (see question 4) and nor in view of the used acceptor (see question 5), at least some side by side comparison should be performed to draw conclusions on the choice of FPs. As such, because of the limited scope, the current study may contribute to confusion without thoroughly discussing the possible downsides in cells and possible advantages of other pairs. Based on this study the readership of Biosensors could choose suboptimal FRET couples for their future work. The rationale should not be so much about using the RFP as acceptor, this is already known from the Mastop study, the rationale should much more be about the particular choice for the RFP, especially within the cellular context.
3) The donor photobleaching issue is not satisfactory addressed: no quantitative comparison is made (Q4).
4) The authors agree that photochomicity of mRuby3 indeed is an issue. In the new manuscript a complete new section (line 305 and further) has been added in the discussion to address the issue. They also mention “initial inconsistencies” in their spectral recordings due to the photochromism, and they now use this confounding issue to explain their aberrant spectral parameters measured for the novel mCer3-mRuby3 FRET pair. In this section they also state that “the observed FRET efficiencies with our different protein pairs do not at all correlate with the calculated Förster radii…”. Again this is confusing and not helping to make a case for this pair, these inconsistencies should be resolved or the data should not be used. The explanation of altered orientation factors is not substantiated by data and very unlikely in view of the flexible linkers employed between donor and acceptor. The issue concerning slow and variable maturation is not addressed, this may not be much of an issue for purified proteins from bacteria, however in cells it is a major concern. (Q5)
5) This issue is not adequately addressed/discussed, it is still unclear what will happen in vivo. (Q6)
6) This issue is not adequately addressed (see also issue 1). (Q7)
7) This issue is not adequately addressed: moreover they do not have mRuby2 “in house”. (Q8)
8) The answer to issue 9) as well as the final lines (354-357) of the discussion (our next step is to test suitability in live cells) is unsatisfactory. Currently only in vitro data are shown. For in vitro data autofluorescence is absent and large spectral overlaps are no problem at all since various unmixing strategies can be readily applied. For in vitro studies there is no reason at all to substitute CFP_YFP for mCerulean3_mRuby3. (Q9)
9) The potential problem of bystander FRET is acknowledged by the authors. No data is presented to address the issue. Because mRuby3 does not behave as a monomer in the OSER assay is a major concern when used in cells. In that respect mRuby3 is certainly no advantage over YFP. (Q10)
10) The issue that the microplate reader has too large bandpasses and is spectroscopically below standards is acknowledged, it is however unclear now what the effect is of this when doing quantitative analysis based on more blurred spectra. (Q11)
11) The issue of Trypsin controls is adequately addressed. Yet the answer of selective donor adsorption to the wall of the employed microtiter plates is very worrying. For steady state FRET techniques this greatly affects calculated FRET efficiencies and donor quenching percentages. (Q12)
12) This issue remains. In fact the explanation may be adsorption of CFP to the wall of the microtiter plate during digestion. In that case the spectroscopic ratio change is underestimated for the CFP-YFP pair and if corrected, it could even imply that in terms of ratio change, the CFP-YFP pair would be a better choice than the mCerulean3-mruby3 pair. (Q13)
13) This is addressed. (Q14)
14) It is acknowledged that the control digestion was not done for the other pairs because for these pairs the trypsin resistant pairs were not even made.
15) It is acknowledged that this is a pilot study, I therefore doubt if the current study is mature enough to merit publication in biosensors, it could however be improved and interesting for the readership.
Author Response
We thank the reviewer for the effort spend in judging our manuscript and for a number of suggestions/comments that definitely help with our future work. In general, we see that this reviewer is not in favour of our manuscript and, of course, respect his opinion, with which we, however, disagree.
We answer to his comments in the following:
The authors have improved the manuscript, however there are still several issues that are not adequately addressed or discussed. What really worries me is that the authors agree with my statement that this is merely a pilot study, and therefore not a matured study. My main concern is that the study is limited in the choice of donor-acceptor pairs, furthermore the extrapolation to cells has not been explored or tested at all for their particular sensor and application. Any conclusion or statement concerning cells can unfortunately not supported by data. My other concern is that without testing this particular donor-acceptor pair in vivo in cells, other scientists may think that this is the best choice, while it is known that in particular mRuby3 does not have the best and favorable properties in cells (photochromicity, extremely slow and variable maturation and clustering issues). To some extent these properties are not an issue in vitro, or otherwise stated, it is fine to perform a spectroscopic study on any pair, but one needs to be very diligent with suggestions that it would also be a good pair within a cellular in vivo context.
It is our impression that this reviewer, who has placed many thought- and useful comments (a big thank you for those), defines goals of our manuscript other than those that we phrase and then concludes that we have not met them. It is obvious that we cannot produce data in live cells in the time frame that is available for this revision. We are not sure inhowfar this reviewer is aware of that. It is perfectly clear that we aim to use our proposed FRET pair in a FAMPIR 2.0 version, yet it will take a few more months to produce publication quality results. The data we have so far are all in-vitro, which is indicated in the title of the manuscript. Our claim that an RFP produces much less interference with cellular autofluorescence comes from simple observations with mRuby3-only-expressing HEK cells that simply would not fit in the manuscript (for (a) being the only cell-based data and (b) being simply a bit trivial). All the rest is based on our experimental data presented in the manuscript. The term "pilot study", in our eyes, does not mean an immature study but a study intended to explore basic conditions for the usefullness of a specific tool.
0) The authors added spectroscopical characteristics, however the extinction coefficients significantly differ from published values. For instance, for mRuby3 they tabulate an EC=79 M-1cm-1, whereas Lin and colleagues (their ref [6]) determined an almost twice higher value: ECs=128 M-1cm-1. If the current study is published this leads to confusion in the literature.
We have repeated the measurements several times, including those for N-terminal variants of mRuby3 that we produced for other purposes. All of these measurements are witihin the range reported by us (in a range up to 10% lower, none are higher). In our view, publishing data that are not identical to those reported by others does not lead to confusion but rather triggers the awareness that data reported in the primary literature are not necessarily absolutely correct (of course, we believe in our data, but I am sure that Li et al believe in theirs, likely for similarly good reasons).
In the discussion (line 306-308) they also underline that the spectral characteristics of mRuby3 indeed differs in their hands as compared to the original study. They attribute that to photochromism (yielding a substantially reduced Förster radius of 6.1 nm as compared to the previously published 6.5 nm) (see also point 2 and 4).
The inclusion of the photochromism in our Discussion was a tribute to this reviewer. We do not see any sign of Ruby photochromism in our experimental data, so far. The photochromism observation was reported by Bindels et al., using quite high radiation energy in confocal microscopy (actually beyond those used for photobleaching in the same study).
Furthermore, the authors refer to figure 4B for enhanced contrast of mCerulean3-mRuby3 as compared to CFP_YFP, yet the data in fig 4B hardly shows a difference in ratio contrast between the two pairs. In addition, in figure 4A, the sensitized emission in case of CFP_YFP is clearly significantly enhanced as compared to the Clover3_mRuby3 and Cerulean3_mRuby3 pairs. So again their point of enhanced contrast is not shown by their data. (see also point 13). We disagree with that: as discussed, the admittedly impressive sensitized emission signal for YFP is riding on a substantial CFP background signal. This is a problem for our FAMPIR analysis because cellular background fluorescence in the wavelength range of the CFP signal shows extremely broad interidividual difference on the single cell level, making spectral separation (bleedthrough correction) very difficult. Of notice is that the authors also admit they do not present data on the lack of autofluorescence in cells and even state their cell data are “absolutely preliminary”. See above
1) Förster radii were incorporated, but the R0 values tabulated for mClover3mRuby3 of 6.1 nm is 0.4 nm lower than the published value. This is due to our lower EC value for mRuby3.
2) The answer to question 3) is not satisfactory: no simple comparison of straightforward other cyan-Red FRET pairs have been presented whereas this is a trivial effort to do so. As stated before: a much more extensive study is published elsewhere (their ref [21]) and the current paper fails to add anything to that published paper. No single convincing case can be made that mCerulean3-mRuby3 is particular beneficial within the cellular context. Not necessarily in view of the donor (see question 4) and nor in view of the used acceptor (see question 5), at least some side by side comparison should be performed to draw conclusions on the choice of FPs. As such, because of the limited scope, the current study may contribute to confusion without thoroughly discussing the possible downsides in cells and possible advantages of other pairs. Based on this study the readership of Biosensors could choose suboptimal FRET couples for their future work. The rationale should not be so much about using the RFP as acceptor, this is already known from the Mastop study, the rationale should much more be about the particular choice for the RFP, especially within the cellular context.
Our study started before the Mastop paper was published. Our approach was to analyze purified fusion proteins in order to first see their behaviour in the absence of a cellular context that can bring in additional factors that complicate the picture. We do not understand why the reviewer is against the publication of a second, independent study to look at the suitability of the mCerulean3_mRuby3 pair, admittedly close to the mTurquoise2_mRuby2 pair in the Mastop study, under different aspects.
3) The donor photobleaching issue is not satisfactory addressed: no quantitative comparison is made (Q4).
Because we did not use mTurquoise2 we have not made a comparison. In our hands, i.e. with our experimental setup we do not observe photobleaching with mCerulean3, nor with any of the other fluorescent proteins employed in our study. This can be attributed to the low total radiation energy resulting from the pulsed light emission in our plate reader. These conditions are well comparable to those in a FACS reading, the method used in FAMPIR, the procedure we plan to use with intact cells.
4) The authors agree that photochomicity of mRuby3 indeed is an issue. In the new manuscript a complete new section (line 305 and further) has been added in the discussion to address the issue. They also mention “initial inconsistencies” in their spectral recordings due to the photochromism, and they now use this confounding issue to explain their aberrant spectral parameters measured for the novel mCer3-mRuby3 FRET pair.
We ourselves do not observe photochromicity with Ruby, see the above comment
In this section they also state that “the observed FRET efficiencies with our different protein pairs do not at all correlate with the calculated Förster radii…”. Again this is confusing and not helping to make a case for this pair, these inconsistencies should be resolved or the data should not be used. The explanation of altered orientation factors is not substantiated by data and very unlikely in view of the flexible linkers employed between donor and acceptor.
The orientation factor is actually the only variable in the formula that is poorly characterized, and therefore the best candidate for an explanation of the phenomenon. Due to the dimerization tendency of the CFP_YFP pair used by us we believe that the explanation is plausible. Alternatively, our Förster radii calculations could be wrong, but here the major "insecurity" is arguably the EC for Ruby, which differs between our own findings and those reported by others. It would change in parallel for both, the Cerulean_Ruby as well as for the Clover_Ruby pair and therefore not change their ratio.
The issue concerning slow and variable maturation is not addressed, this may not be much of an issue for purified proteins from bacteria, however in cells it is a major concern.
(Q5) No issue for the present analyses
5) This issue is not adequately addressed/discussed, it is still unclear what will happen in vivo. (Q6) This is an issue we need to address once we express the protein in cells. However, in contrast to the OSER assay, Ruby will be expressed intraluminally in the ER in our FAMPIR approach, and therefore the formation of whorls as observed in the OSER assay that result from the interaction of proteins expressed on the cytoplasmic face of the ER are a priory not expected.
6) This issue is not adequately addressed (see also issue 1). (Q7)
See our initial comment
7) This issue is not adequately addressed: moreover they do not have mRuby2 “in house”. (Q8)
We do not understand the remaining issue. Apparently, the reviewer is in the luxury position to have many fluorescent proteins at hand. Inspired by the discussion, we have just acquired a few new cDNAs but first need to clone and express them to make further comparisons.
8) The answer to issue 9) as well as the final lines (354-357) of the discussion (our next step is to test suitability in live cells) is unsatisfactory. Currently only in vitro data are shown. For in vitro data autofluorescence is absent and large spectral overlaps are no problem at all since various unmixing strategies can be readily applied. For in vitro studies there is no reason at all to substitute CFP_YFP for mCerulean3_mRuby3. (Q9)
It is obvious that this reviewer would like to see in cell data that we simply do not have at this timepoint. Our study is designated an in vitro analysis.
9) The potential problem of bystander FRET is acknowledged by the authors. No data is presented to address the issue. Because mRuby3 does not behave as a monomer in the OSER assay is a major concern when used in cells. In that respect mRuby3 is certainly no advantage over YFP. (Q10)
We have no indication for dimer/oligomer formation of mRuby3 during size exclusion chromatography but will certainly monitor the behaviour of mRuby3 when going into living cells. We have a negative interaction control (membrane anchor only) in our FAMPIR assay to address this potential issue.
10) The issue that the microplate reader has too large bandpasses and is spectroscopically below standards is acknowledged, it is however unclear now what the effect is of this when doing quantitative analysis based on more blurred spectra. (Q11)
Maybe we are a bit too defensive about our microplate reader. It is not a high end fluorimeter but a work horse in high throughput analysis of fluorescent samples. It uses two tandem monochromators, ie. two prisms serially connected to select the desired wavelength, for both the excitation and the emission wavelengths. There are no cut-off filters in use. Actually, the machine is able to measure FRET in live cells in the microplate format. If we find the expected increase in sensitivity we will try this option as an alternative to the FACS approach. Thus, we regard it as an advantage to be able to run the initial analyses on this machine.
11) The issue of Trypsin controls is adequately addressed. Yet the answer of selective donor adsorption to the wall of the employed microtiter plates is very worrying. For steady state FRET techniques this greatly affects calculated FRET efficiencies and donor quenching percentages. (Q12)
In a series of experiments we find the absorption of CFP, as a single protein or as a YFP fusion, to amount to 25% during the observation period and to plateau after 30 min at 33%. This is certainly not ideal but does not quantitatively explain the unexpected lack of donor dequenching on trypsin digestion. (As discussed, given the substnatuial FRET efficiency, a much larger increase would be expected). We therefore still regard this issue as unresolved and present this as an open question in the discussion.
12) This issue remains. In fact the explanation may be adsorption of CFP to the wall of the microtiter plate during digestion. In that case the spectroscopic ratio change is underestimated for the CFP-YFP pair and if corrected, it could even imply that in terms of ratio change, the CFP-YFP pair would be a better choice than the mCerulean3-mruby3 pair. (Q13)
13) This is addressed. (Q14)
14) It is acknowledged that the control digestion was not done for the other pairs because for these pairs the trypsin resistant pairs were not even made.
15) It is acknowledged that this is a pilot study, I therefore doubt if the current study is mature enough to merit publication in biosensors, it could however be improved and interesting for the readership.
See our initial comment
Reviewer 2 Report
The paper can now be accepted
Author Response
We cordially thank the reviewer for his constructive previous comments and for his positive vote!